# Strain-Field Modifications in the Surroundings of Impact Damage of Carbon/Epoxy Laminate

**DOI:** 10.3390/polym14163243

**Published:** 2022-08-09

**Authors:** Jarmil Vlach, Radek Doubrava, Roman Růžek, Jan Raška, Jan Horňas, Martin Kadlec

**Affiliations:** VZLU—Czech Aerospace Research Centre, 19900 Prague, Czech Republic

**Keywords:** composite, laminate, impact, experiment, simulation, correlation, structural analysis, strain-field distribution, Abaqus

## Abstract

The relationship between deformation and stress is crucial for any elasto-plastic body. This paper deals with the experimental identification of the basic parameters of the composite laminate model in relation to the finite element model. Standardized tensile, impact, and post-impact tests on a carbon fiber-reinforced epoxy laminate were used. The method by which the elasticity and failure parameters were obtained from the initial components is described. In the article, the modes of initiation and complete failure of samples in tensile tests, which are compared with the simulation, are presented. Furthermore, the article deals with the issue of the generation and detection of damage by low-speed impact, which can be caused by contact with moving objects, due to improper handling or maintenance. The results of impact analysis simulations are shown in the context of strain-field distribution changes obtained with the help of digital image correlation. The results showed high agreement between the calculations and the experiments. Based on this agreement, simulations of impact damage for various energies were performed. These simulations were used to determine the approximate sizes of the affected zones in relation to the impact energy. The results are finally discussed in the context of the possible use of structural health monitoring based on strain modifications.

## 1. Introduction

Polymer composites are used in the aerospace industry due to the high strength-to-weight ratio [1]. Low-velocity impact damage of carbon-fiber reinforced polymer laminates is a serious damage mechanism that limits the performance of the material [2]. This type of impact damage can occur during the service of a structure (runway debris, hails, manipulation around aircraft during loading and maintenance) or by handling during manufacturing. The damage causes surface indentation (dent) and below-surface damage such as matrix cracking, fiber breakage, and delamination. Visibility of the damage is based on energy and impact tip radius [3]. The effect of this damage is significant especially for compression strength, which is reduced significantly due to internal damage that cannot be detected by visual examination [4]. During the loading of a damaged laminate, the strain field is affected around the impact damage [5]. This strain field modification can be used to detect the impact damage by static strain sensors such as strain gauges or fiber Bragg grating (FBG) sensors in an optical fiber. Readings of a network of surface sensors can be input into an artificial neural network that will evaluate the damage location by recognition of a pattern in the strain change. This technique can be used for structure health monitoring (SHM). The results of the virtual testing of strain field changes for different impact energies and different locations can be used to train the neural network and enhance the state-of-art of SHM systems, which are based mostly on a single sensor response without any sensor network interaction.

The issue of structural and stress analysis, which emphasizes the accuracy of solved problems, is always based on the mutual correlation of calculations with experiments. These are related to the relationship between body deformation and applied load [6,7,8]. This property is essential for any elastic-plastic body or machine assembly having a similar character. If the problem of predicting the existence of such a body in the context of physical reality is solved, it must be assumed that there is a certain prediction model related to the reference states.

The present paper deals with the correlation of a series of experiments and a finite element model (FEM) of a composite board in relation to specifically determined load cases [9]. These condition changes were investigated through standardized tests before and after the violation to provide reliable learning data for machine learning devices. Digital optical correlation (DIC) methods were used to analyze deformations and relative deformations of test specimens [10,11].

There is the possibility to divide the response of each material to loading stimuli in a rough way into three intervals, which form an inseparable whole when assessing safety and durability. They are the intervals of elasticity, plasticity, and the breaking interval, which is beyond the strength of the material [12,13]. The compilation of a valid computational model, with respect to the consequences of the failure, represents balanced weights only if the phases of its construction run in parallel with the experiments within predetermined limits of accuracy [14]. Once the model is validated, it is no longer necessary to perform many additional experiments. The consequences of the load cases, which are between the limit states, can then be calculated or interpolated between the FE calculations.

The first presented and best conditioned technological test, by means of which the basic material characteristics can be reliably identified, is the tensile test [15]. This is mainly because the jaws of the testing machine are very rigid. By clamping the sample to the jaw, all but one of the degrees of freedom is removed and monitored. This ensures that the experiment has a minimal effect of possible play or inaccuracies. This is essential for possible model validation, as this may be different for other tests. It is well known from practice that even ordinary linear solvers in the field of small deformations reach high accordance with this experiment, which results from the above context.

The second presented test, which is damage caused by collisions with small objects, is the impact test without perforation of the sample wall [16]. The test is suitable for the identification of the cohesion parameters of individual layers and the degree of absorption of mechanical energy. From the starting height of the normalized impactor and its rebound, the initial and final energy states of the system of bodies can be clearly determined.

The last test that the paper deals with is the tensile test after the impact failure. This test was performed to determine changes in the displacement field and the relative deformations of the sample around the impact damage. The DIC method was used to analyze the displacement field change. This is because local failure without wall perforation leads to only small changes in stiffness and cannot be directly demonstrated by a simple experiment on a sample or component.

By correlating these tests and comparing them, we have created a computational model by which we can calculate the approximate extent of damage between the extreme points of the test interval. Such a model can help to generate learning data for automated evaluation by advanced statistical methods such as neural networks. The job overview is shown in Figure 1.

## 2. Methodology

### 2.1. Material Description

All experiments were performed on a 12-layer laminate with the composition (−45, +45, 0, 90, 0, 0)_S_ with total thickness of 1.584 mm. The laminate was made of a unidirectional carbon fabric (Toray, Tacoma, WA, USA) in an epoxy matrix (Hexcel, Saronno, Italy) with a volume fraction of 53.3%. The plies were hand laminated, vacuum bagging was performed, and the laminate was cured in an autoclave at 180 °C and 6 bar pressure. Fibers were of a non-commercial nature provided only for a specific scientific research purpose; the mechanical properties of the fibers can be considered equivalent to the T700G standard [17]. The properties of the matrix correspond to epoxy 3501-6 [18]. The scheme of the laminate composition is shown in Figure 2.

### 2.2. Tensile Test

Tensile tests were performed according to ASTM D3039 [15] on 250-mm-long test specimens with a mean cross-sectional area of 198 mm^2^. During the experiment, the dependence of the force on the displacement of the sample was checked on an Instron 55R1185 testing machine (Instron, Norwood, MA, USA). Longitudinal elongation and transverse narrowing were measured with an Epsilon biaxial extensometer (Epsilon Technology Corp, Jackson, WY, USA). The sample was loaded to break in the absence of an extensometer. The experiment and the dimensions of the test specimen are shown in Figure 3a,b.

### 2.3. Impact Test

Impact tests were performed according to ASTM D7316 [16]. The implementation of the experiment is shown in Figure 4a, and the dimensions of the sample are shown in Figure 4b.

The impact test was performed for various energies to check the extent of damage and penetration of the impactor into the sample. The sample was impacted in the geometric center with a hemispherical test specimen with a diameter of 12.7 mm. The deformation of the sample at the impact site was measured with an optoNCDT ILD 2300-20 laser interferometer [19] (Micro-Epsilon Messtechnik, Ortenburg, Germany). The highest considered kinetic energy of the presented comparative test was 10.88 J. During the experiment, the height of the impactor, the rate of impact, and the amount of deformation (δ) of the sample at the location below the impactor were checked. After the impact, the permanent deformation caused by the penetration of the test specimen was measured. The parameters were compared with the result of the simulations. The evaluation scheme is shown in Figure 5a,b. Points A and A’ represent the area of the specimen before and after deflection, which was checked by laser beam.

### 2.4. Post-Impact Tensile Test

After the verification of the basic parameters, a post-impact tensile test was performed to check the stress distribution field in the surroundings of the affected zone. The experiment (Figure 6a) was performed in a similar way as the initial tensile test but without the presence of a biaxial extensometer (Epsilon Technology Corp, Jackson, WY, USA) and on test specimens of non-standard dimensions (Figure 6b), which represent a section of the aircraft structure. The extensometer was replaced by a more complex Digital Optical Correlation (DIC) method [11]. A specific case of the impact of the laminate board is shown in Figure 7a,b.

### 2.5. Tensile Test Simulation

The simulation was performed using the Abaqus/Standard and Explicit 6.14 systems (Dassault Systemes, Vélizy-Villacoublay, France). The FE model was created using the ply-by-ply (PBP) method using SC8R elements with a mean size of 1.5 mm with an emphasis on network homogeneity. Cohesive contact was not considered in the tensile test model; the individual layers were connected by identical nodes. The setting of elastic parameters of the model is shown in Table 1. The same parameters can be determined for a given reinforcement representation by the Mean-Field-Homogenization (MFH) method [20] with the Mori-Tanaka formulation or by the correctly performed representative volume element (RVE) method [21]. The composite failure was considered according to the Hashin scheme [22,23], and its parameters for the single-layer laminate, which were derived by the (RVE) method, are shown in Table 2 and Table 3. A graphical representation of the properties of an intact laminate from which the rate of absorption of deformation energy can be derived is shown in Figure 8a,b. The area of plasticity proved to be insignificant for the given composition, and so it was not considered in the model [24,25].

### 2.6. Impact Test Simulation

Impact simulation had to be supplemented by cohesive contact parameters [7,26,27], which affect the potential for delamination. The contact properties in this case were directly derived from the properties of the laminate, given in Table 4 and Table 5. The table of cohesive contact strengths further shows that the resistance of the layers to separation is considered depending on the mutual orientation.

It is important to highlight that the evolution properties of the damage parameters are extremely sensitive to changes in manufacturing technology and can fundamentally affect the actual resistance of the laminate to detachment as well as the simulation results. Their settings must therefore be monitored and optimized as the conditions change. The impact simulation was performed using the Abaqus/Explicit solver. The use of this solver is particularly suitable for short processes and strongly nonlinear tasks with material failure, stiffness, and contact drop. Recall that consideration of contact is essential in the impact task because delamination results in the division of the original body into multiple bodies.

### 2.7. Post-Impact Simulation Test

After simulating the impact with material failure [28,29,30,31,32,33], the task was linearized, and the results were reformulated for the Abaqus/Standard solver. Linearization was performed using the procedure detailed in [34,35]. Its purpose is to replace several damage modes with a single material card. Individual material cards can be calculated for consolidated damages $d_{1}$, $d_{2}$, and $d_{s}$ according to Table 6. The linearization of the contact is performed in such a way that the nodes in the contact pairs that have been detached are left free, and the network is not interconnected at these places.

## 3. Results and Discussion

### 3.1. Tensile Test Simulation Results

First, the agreement of the dependence of the force response on deformation was checked, which is shown in Figure 9.

To verify the Poisson ratio of the composite laminate, the field of relative deformations is shown in Figure 10. Figure 10a shows the lateral distribution of the longitudinal axial deformations, and Figure 10b shows the distribution of relative deformations in the contraction direction. The results show that the fields of relative deformations are homogeneous in the elastic range 0.001 to 0.003, which is determined by the standard.

The results of the agreement of the model and the quantities, which were verified by the tensile test, are shown in Table 7. Experimentally determined quantities are given with a probability of 95% in the given interval [15,16,36].

The strength parameters were checked in two steps. First, the failure initiation mode of the individual layers was checked, and then the destruction of the model sample was compared with the experiment. The location of the failure initiation is shown in Figure 11a. A comparison of the final failure mode found from the simulation and the experiment is shown in Figure 11b,c.

### 3.2. Impact Test Simulation Results

The comparison of the monitored quantities obtained from the results of the simulation and experiments is shown in Table 7 and Table 8.

A simplified visualization of the damaged bottom and top layers of the laminate as a function of the impact energy is shown in Figure 12 and Figure 13. The damage profile is shown in Figure 14. Reference control states in relation to the experiments are marked in black. The gray-colored pictures show the damage to the material, which is not related to the experiments, but is only determined by calculation.

### 3.3. Post-Impact Simulation Results

A simplified visualization of the damaged bottom and top layers of the laminate as a function of the impact energy is shown in Figure 12 and Figure 13. The damage profile is shown in Figure 13. Reference control states in relation to the experiments are marked in black. Material damage, which is not related to the experiments, is shown in gray. The comparison of the distribution of the main deformations took place at a load of 80 kN (Figure 15 and Figure 16) and 110 kN (Figure 17 and Figure 18). For a clearer presentation, vertical and horizontal paths lying on the axes of symmetry of the samples were selected. The DIC analysis can be considered valid at an interval of approximately 10 mm from the edge of the plate to the edge of the dent caused by the impact.

This is because the dent is significantly curved compared to the sample plane. Due to its deformations, which occur during axial loading, large relative deformations appear during evaluation. However, the deformation distribution around the impact can be used for rough comparison. The obtained waveforms were not intentionally filtered to illustrate the raw results.

Importantly, however, the correlation clearly indicates the limit of the size of the dent that results from permanent deformations. From the results, it can be concluded that the correlation with the calculation with the experiment is high, even though the analysis took place only after the linearization of the results. The agreement in the direction of the horizontal paths of both samples at different loads is especially noticeable. The relative deformations along the vertical path differ visibly, but we do not have a relevant explanation for this difference, although the diameter of the resulting hole can be visibly determined very well from the diagram. However, it can be clearly stated that DIC shows extreme deviations from the environment on a smooth surface, and the simulation provides more realistic information about the actual deviations. However, both methods lead to the identification and localization of damage in an unmistakable way. The results of the strain-field modification for the intact and broken samples are shown in Figure 19 and Figure 20.

### 3.4. Results Extrapolation

As mentioned in the introduction, the first goal of the work was to verify the model in terms of reference states. The second goal was to use a calibrated model to implement the simulation, which would provide data for machine learning measures or allow easy identification of damage.

For this purpose, six simulations of the impact of energy in the range of 0.5 to 10 J were performed. The simplest characteristic of the damaged elements, which were the process of linearization of the grouping, is to quantify their amount. From this, provided that the network is homogenous and each element has the same area, the affected area of the individual layers can be calculated. The sizes of the areas of the damaged zones depending on the impact energy are shown in Figure 21.

In this case, the DIC results can be unambiguously identified with the simulation result. At the same time, it has been shown that we can perform pairing of DIC data for samples without damage and those damaged by impact. The color maps of the strain-field modifications are shown in Figure 20. These directly correlate with experiments from post-impact tensile tests. A comparison of the results obtained with the impact for lower energies has been verified, but due to the large scale, they are not part of this paper.

Overall, it appears as though it is not possible to compare the results absolutely, but only in terms of changes in the applied load. In the middle of the impacted area, the simulation points to strain-field changes in the order of 1000 microstrains, as shown in Figure 19. Any SHM method used should then be in this possibly lower order of detection.

From Figure 16 and Figure 18, the detectability depends on the direction of the evaluation path and the orientation of the top (bottom) layer of the laminate with respect to the direction of loading. In the present case, it would be possible to delimit the minimum detectable area with a circle with a maximum diameter of 40 mm. It is apparent that the size of the detectable zone does not depend on the size of the load. If the load is in the elastic region, the strain modification differs only in magnitude.

It is clear from the damage cross-sections that the damage identification is most effective from the surface of the sample (component). Simulations show that the immediate surroundings of the midplane are only locally affected. In this context, however, it is not possible to generalize completely, because the same oriented layers have a significantly lower tendency to delaminate.

## 4. Conclusions

In this article, the material model of a composite laminate was shown. Its parameters are fully sufficient for impact simulations with the help of the Abaqus/Explicit or Abaqus/Standard system. The credibility of the results was confronted with experiments, but a specific example showed that the impact leads to higher damage on the side of the sample that is not exposed to the impact. In secondary elements of airframes, this kind of damage is usually not a problem, but it points to the special insidiousness of the phenomenon of poorly visible impact and the necessity of its identification in the primary constructions. Although the damage from barely visible impact damage usually does not tend to spread, it can lead to a decrease in static stiffness and strength. The simulations further show that the damage rate is layer-dependent in the laminate. It can be seen from the results that the failure of the layers in the vicinity of the mid-plane is lower than the failure rate on the surface. It should be emphasized that the model results were only validated in the presented rate and damage of the laminate. Another laminate with a different composition may differ from the presented one. In general, the material model can therefore be used as a reference for laminates with the same or very similar properties. For future research, the methodology is planned to be validated on the same laminate with 24 layers to study the stiffness parameters. The most important findings have been summarized in individual points shown below.

The basic experiments and their relation to the parameters of the FE model were shown.A high correlation with partial results of reference calculations with experiments was shown.The results of partial experiments of tensile and impact tests fall within the confidence intervals with a probability of 95%.The simulation showed that the dependence of the increment of the area affected by the impact is nonlinear.A method of identifying the extent of damage by the DIC method in relation to the impact energy of the test specimen and changes in relative deformations around the damage was successfully presented.Simulations have confirmed that areas near the top and bottom layers are most effective for collecting strain-field data changes. The possible application of SHM should be focused on these cross-sectional areas.Correlated models will be used to generate strain-field data in relation to the presented load cases.

## Figures and Tables

**Figure 1 polymers-14-03243-f001:**
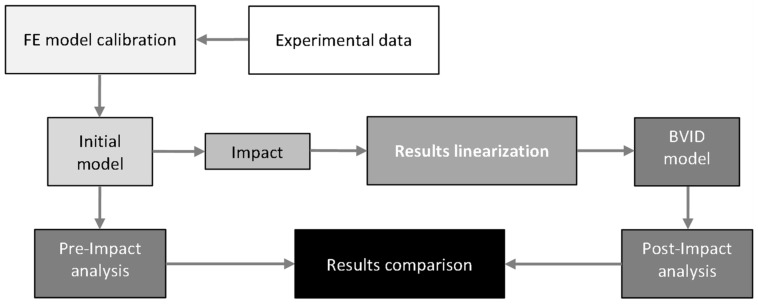
Workflow schema.

**Figure 2 polymers-14-03243-f002:**
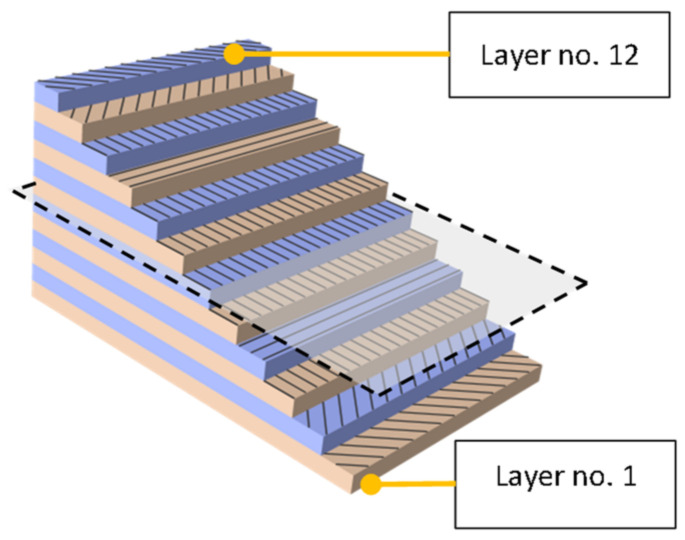
Ply-by-ply model description.

**Figure 3 polymers-14-03243-f003:**
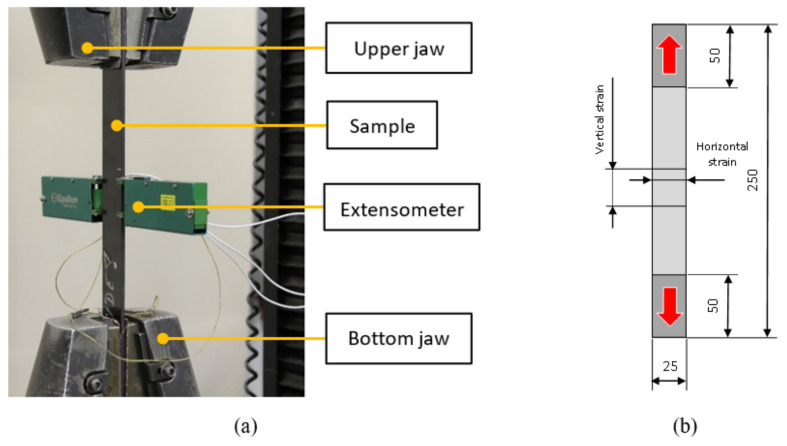
Tensile test; Experiment (**a**); Tensile sample (**b**).

**Figure 4 polymers-14-03243-f004:**
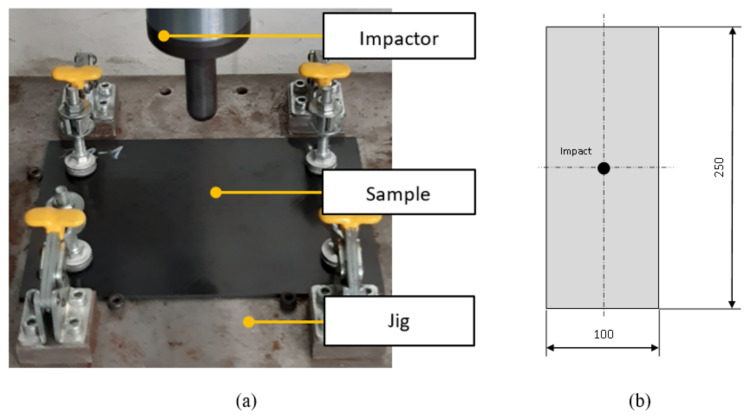
Impact on the specimen; Experiment (**a**); Impact sample (**b**).

**Figure 5 polymers-14-03243-f005:**
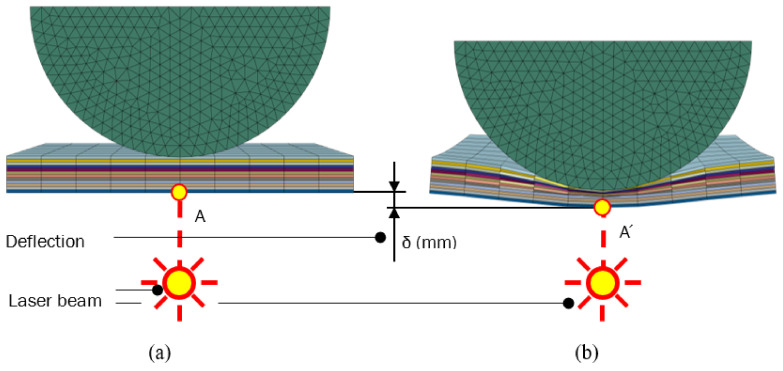
Deflection under the impactor from the simulation results; Initial state of impact (**a**); Maximum plate displacement under impactor (**b**).

**Figure 6 polymers-14-03243-f006:**
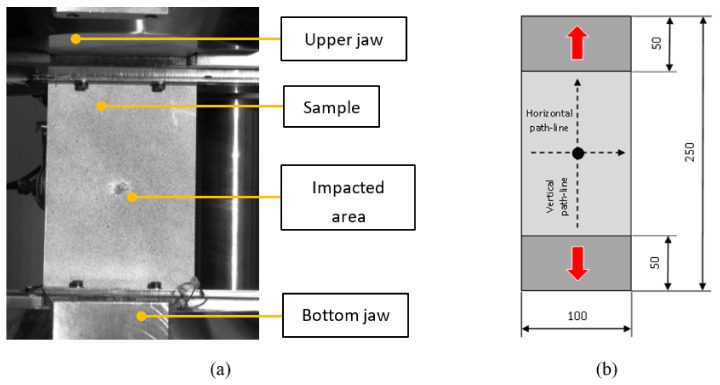
Post-impact tensile test; Experiment (**a**); Sample with paths and dimensions (**b**).

**Figure 7 polymers-14-03243-f007:**
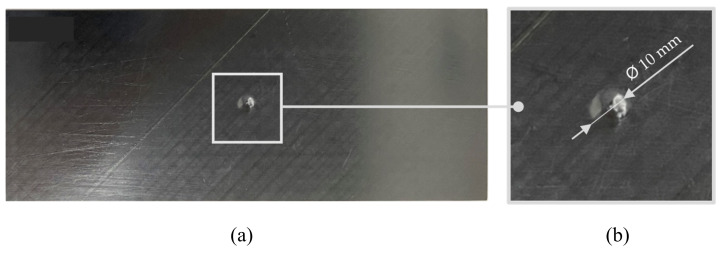
Specimen after impact E = 10 J; Overall specimen view (**a**); Detailed view (**b**).

**Figure 8 polymers-14-03243-f008:**
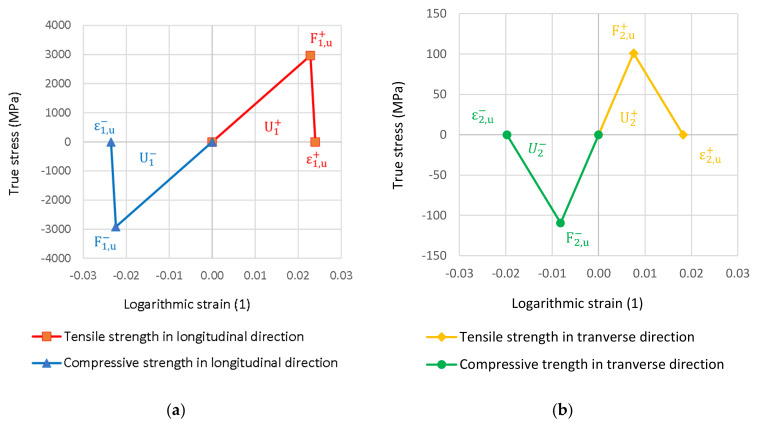
Hashin damage diagram in longitudinal (**a**) and transverse laminate (**b**) direction.

**Figure 9 polymers-14-03243-f009:**
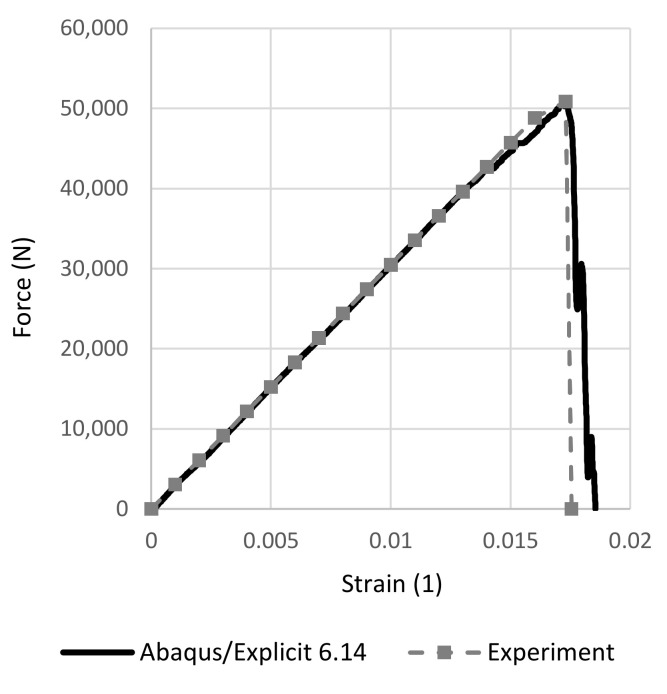
Force–strain FEM tensile sample response.

**Figure 10 polymers-14-03243-f010:**
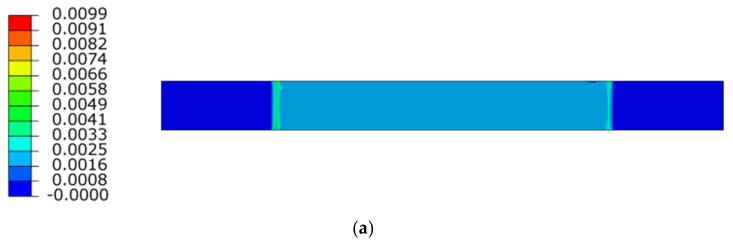
Principal strain field distribution, Vertical (**a**), Horizontal (**b**).

**Figure 11 polymers-14-03243-f011:**
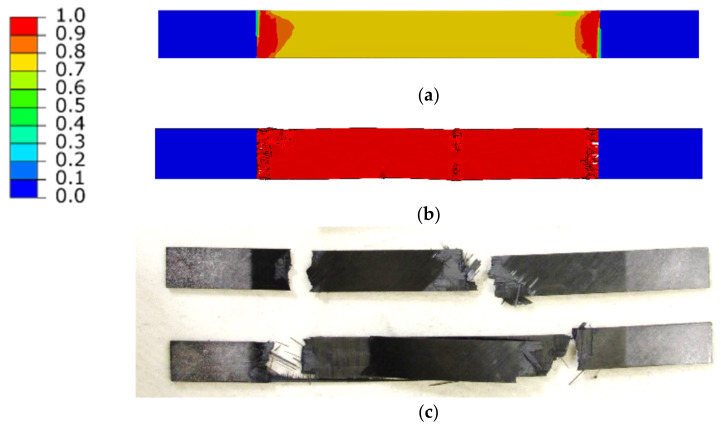
Damage level of the specimen; Crack initiation from the FE model (**a**), Final damage from the FE model (**b**), Damage from the experiment (**c**).

**Figure 12 polymers-14-03243-f012:**
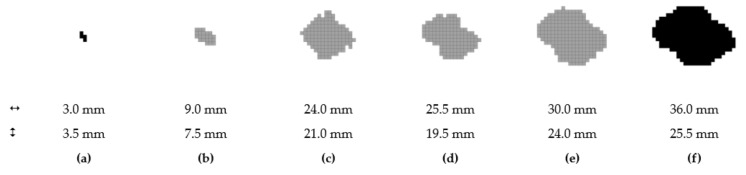
Damaged area at layer no. 1; 0.5 J (**a**); 1.0 J (**b**); 2.5 J (**c**); 5.0 J (**d**); 7.5 J (**e**); 10.0 J (**f**).

**Figure 13 polymers-14-03243-f013:**
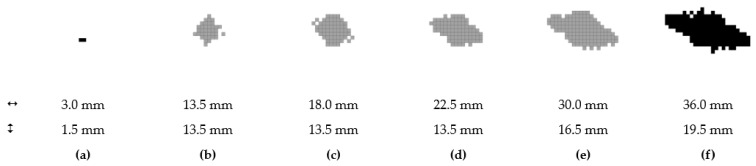
Area damaged from impact; layer no. 12; 0.5 J (**a**); 1.0 J (**b**); 2.5 J (**c**); 5.0 J (**d**); 7.5 J (**e**); 10.0 J (**f**).

**Figure 14 polymers-14-03243-f014:**
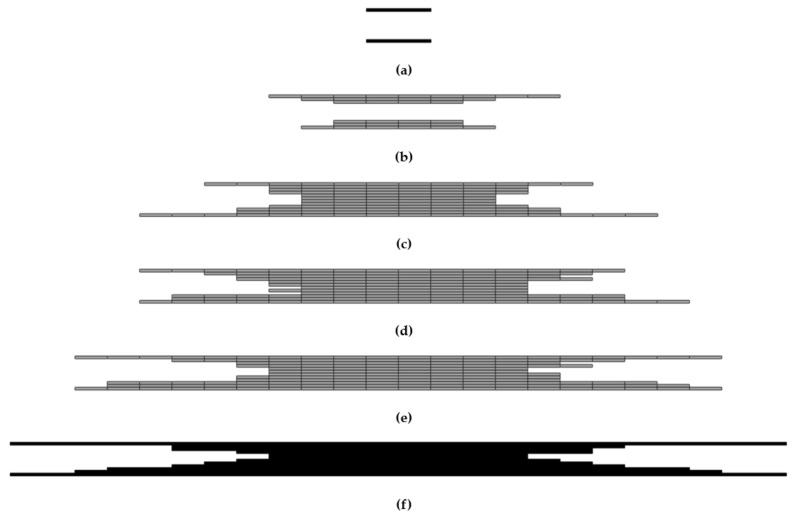
The cross-section profile of the damaged area; 0.5 J (**a**); 1.0 J (**b**); 2.5 J (**c**); 5.0 J (**d**); 7.5 J (**e**); 10.0 J (**f**).

**Figure 15 polymers-14-03243-f015:**
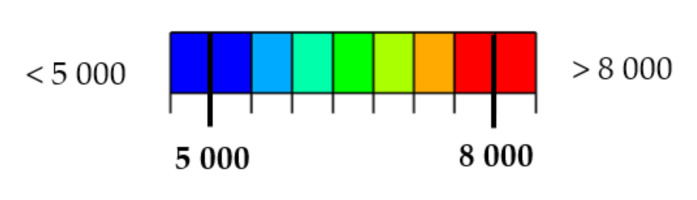
Max. principal strain distribution (μm/m) at load 80 kN; (**a**) Simulation; (**b**) Experiment sample 1; (**c**) Experiment sample 2.

**Figure 16 polymers-14-03243-f016:**
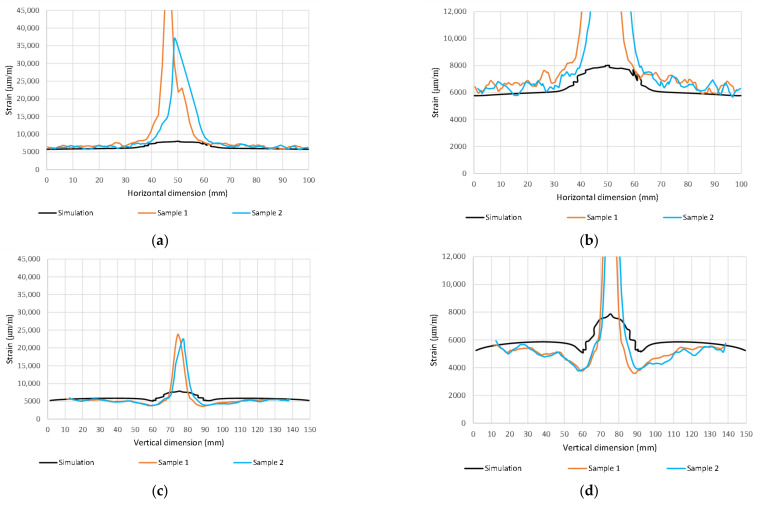
Max. principal strain distribution (μm/m) at load 80 kN; (**a**) Horizontal path—scale 1; (**b**) Horizontal path—scale 2; (**c**) Vertical path—scale 1; (**d**) Vertical path—scale 2.

**Figure 17 polymers-14-03243-f017:**
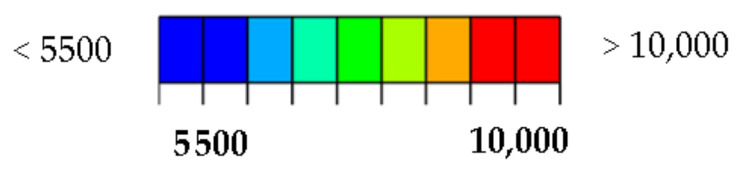
Max. principal strain distribution (μm/m) at load 110 kN; (**a**) Simulation; (**b**) Experiment sample 1; (**c**) Experiment sample 2.

**Figure 18 polymers-14-03243-f018:**
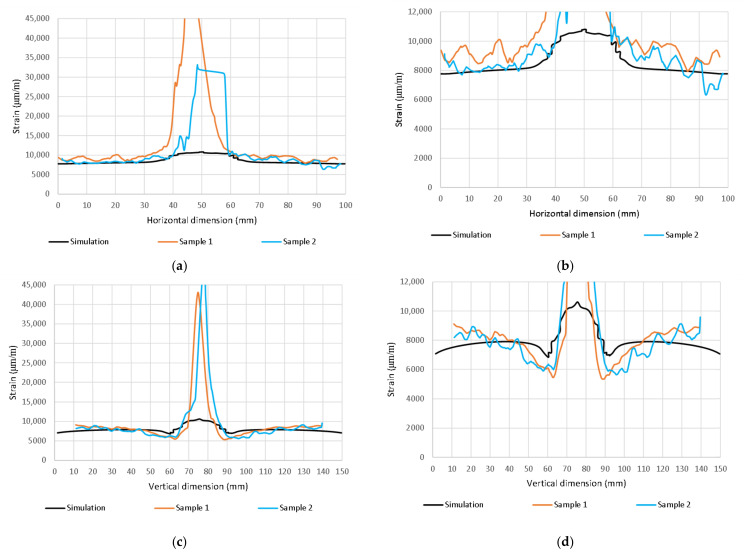
Max. principal strain distribution (μm/m) at load 110 kN; (**a**) Horizontal path—scale 1; (**b**) Horizontal path—scale 2; (**c**) Vertical path—scale 1; (**d**) Vertical path—scale 2.

**Figure 19 polymers-14-03243-f019:**
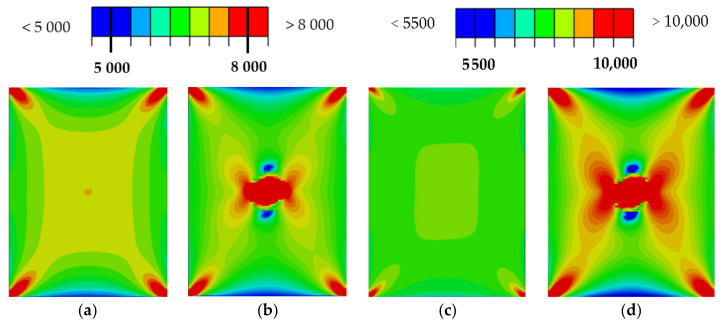
Max. principal strain distribution (μm/m); Pristine 80 kN (**a**); Damaged (10 J) 80 kN (**b**); Pristine 110 kN (**c**); Damaged (10 J) 110 kN (**d**).

**Figure 20 polymers-14-03243-f020:**
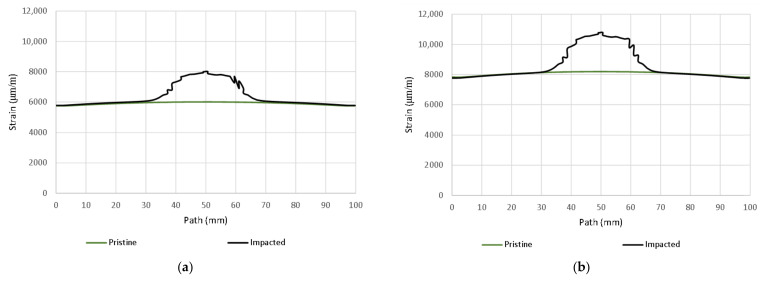
Comparison of max. principal strain distribution (μm/m); (**a**) Horizontal path at load 80 kN; (**b**) Horizontal path at load 110 kN, (**c**) Vertical path at load 80 kN; (**d**) Vertical path at load 110 kN.

**Figure 21 polymers-14-03243-f021:**
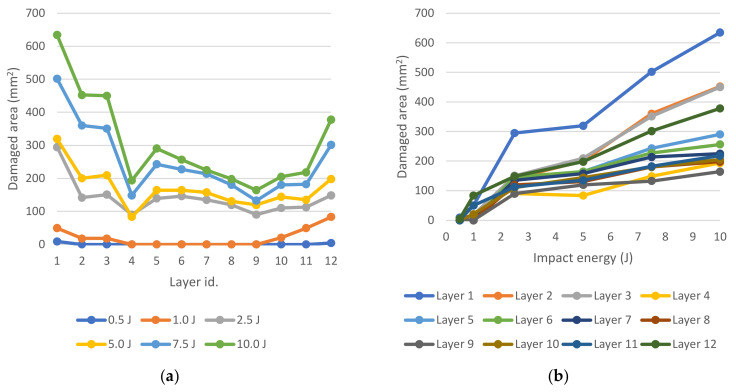
Ply damage distribution; Damaged area vs. layer (**a**); Damaged area vs. impact energy (**b**).

**Table 1 polymers-14-03243-t001:** The elastic properties of the lamina.

$E_{1}^{0}(\mathrm{MPa})$	$E_{2}^{0}(\mathrm{MPa})$	$\nu_{12}^{0}(1)$	$G_{12}^{0}(\mathrm{MPa})$	$G_{13}^{0}(\mathrm{MPa})$	$G_{23}^{0}(\mathrm{MPa})$
129,840	13,340	0.26	4890	4890	4630

**Table 2 polymers-14-03243-t002:** The strength of the lamina.

$F_{1,u}^{+}(\mathrm{MPa})$	$F_{1,u}^{-}(\mathrm{MPa})$	$F_{2,u}^{+}(\mathrm{MPa})$	$F_{2,u}^{-}(\mathrm{MPa})$	$F_{12,u}\;\left(F_{13,u}\right)(\mathrm{MPa})$	$F_{23,u}(\mathrm{MPa})$
2965.41	2911.81	100.88	109.42	100.76	98.41

**Table 3 polymers-14-03243-t003:** The Hashin damage evolution parameters of the lamina.

$U_{1}^{+}\;(\mathrm{mJ}/{\mathrm{mm}}^{2})$	$U_{1}^{-}\;(\mathrm{mJ}/{\mathrm{mm}}^{2})$	$U_{2}^{+}\;(\mathrm{mJ}/{\mathrm{mm}}^{2})$	$U_{2}^{-}\;(\mathrm{mJ}/{\mathrm{mm}}^{2})$
35.56	34.28	0.92	1.08

**Table 4 polymers-14-03243-t004:** The strength parameters of cohesive contact.

Φ	$R_{1,u}(\mathrm{MPa})$	$R_{13,u}(\mathrm{MPa})$	$R_{23,u\;}(\mathrm{MPa})$
>45	100.88	100.76	100.76
≤45	174.03	173.44	173.44

**Table 5 polymers-14-03243-t005:** The damage evolution parameters of cohesive contact.

$D_{1}\;(\mathrm{mJ}/{\mathrm{mm}}^{2})$	$D_{13}\;(\mathrm{mJ}/{\mathrm{mm}}^{2})$	$D_{23}\;(\mathrm{mJ}/{\mathrm{mm}}^{2})$
35.56	0.92	0.92

**Table 6 polymers-14-03243-t006:** Proper elastic properties of damaged lamina.

$E_{1}^{d}(\mathrm{MPa})$	$E_{2}^{d}(\mathrm{MPa})$	$\nu_{12}^{d}(1)$	$G_{23}^{d}(\mathrm{MPa})$	$G_{13}^{d}(\mathrm{MPa})$	$G_{12}^{d}(\mathrm{MPa})$
$E_{1}^{0}\left(1-d_{1}\right)$	$E_{2}^{0}\left(1-d_{2}\right)$	$\nu_{12}^{0}\left(1-d_{1}\right)$	$G_{12}^{0}\left(1-d_{s}\right)$	$G_{12}^{0}\left(1-d_{s}\right)$	$G_{12}^{0}\left(1-d_{s}\right)$

**Table 7 polymers-14-03243-t007:** Sample stiffness and strength check (0°).

	Experiment	Simulation
Tensile modulus	76,965 ± 863 MPa	76,964 MPa
Poisson ratio	0.30 ± 0.03	0.3
Ultimate tensile load	49,820 ± 1075 N	50,854 N

**Table 8 polymers-14-03243-t008:** The penetration of the sample by impactor.

	Experiment	Simulation
Energy	10.45 ± 0.22 J	10.45 J
Penetration	6.99 ± 0.21 mm	6.71 mm
Residual deformation	1.12 ± 0.36 mm	0.77 mm

## Data Availability

The data presented in this study are available on request from the corresponding author.

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
