# Peer review of "Strain-Field Modifications in the Surroundings of Impact Damage of Carbon/Epoxy Laminate"

_polymers, 2022, doi:10.3390/polym14163243_

Round 1

Reviewer 1 Report

The article is interesting and, importantly, it touches upon the issue of the generation and detection of damage by lowspeed impact. The presented results showed high agreement between the calculations and the experiments, which is a confirmation of the well-conducted research. I recommend minor revision before publication.

Comments

-In the abstract, please add information about the composites tested (epoxy with carbon fiber) and the method of production, because it is of great importance when analyzing the results.

-In introduction, there is no clearly defined purpose of the work in relation to the state of the art.

-In point 2. Methodology, no information, what method was laminate made of, manual, vacuum or pressing?

-Please explain the abbreviations used in tables 1, 2 and 3, eg ??? ?? ???? ??? ?

- In tables 1, 2, 3, there is also no indication of measurement uncertainty +/-

-Likewise, in Tables 4, 5 and 8, there is no explanation of the abbreviations used. Moreover, table 8 is followed by 5 and where are tables 6 and 7.

- In conclusion, there are no prospects for the future regarding the methodology of the conducted research.

Author Response

Dear reviewer, we thank you for the comments that will improve the article. You can find responses to specific comments below. The manuscript has the same responses in bubbles, so you can easily see the change.

Comments

1) In the abstract, please add information about the composites tested (epoxy with carbon fiber) and the method of production, because it is of great importance when analyzing the results.

Re: We added statement about a carbon fibre-reinforced epoxy laminate to the abstract. The method of production was added to paragraph 2.1.

2) In introduction, there is no clearly defined purpose of the work in relation to the state of the art.

Re: In addition to previous research, which results were published in [9] and [35], this paper was only focused on the model and experiment match. This is the only first stage of research which will lead to advanced SHM system that will enhance the state of art. A short text was added to the introduction.

3) In point 2. Methodology, no information, what method was laminate made of, manual, vacuum or pressing?

Re: The method of production was added to paragraph 2.1.

4) Please explain the abbreviations used in tables 1, 2 and 3, eg ??? ?? ???? ??? ?

Re: The shortcuts are explained in Abbreviations section.

5) In tables 1, 2, 3, there is also no indication of measurement uncertainty +/-

Re: The only elastic property of multiaxial laminate was measured. The elastic properties of lamina were calculated following Mori-Tanaka method. The strength parameters were optimized and calculated using Representative volume element method. The properties of single lamina are only product of optimization process, which is pointed in the text and described in [9] Vlach, J.; Raška, J.; Horňas, J.; Petrusová, L. Impacted area description effect on strength of laminate determined by calculation.

6) Likewise, in Tables 4, 5 and 8, there is no explanation of the abbreviations used. Moreover, table 8 is followed by 5 and where are tables 6 and 7.

Re: The tables were renumbered in proper order. Parameters are referenced in Abbreviations section.

7) In conclusion, there are no prospects for the future regarding the methodology of the conducted research.

Re: The methodology is planed to be validated on the same laminate with 24 layers. We added the plan to the conclusion.

Reviewer 2 Report

General comments:

This paper investigated the strain-field modifications of a CFRP laminate with/without impact damages through experimental and FEA approaches. The results were just reported and summarized without sufficient interpretation, and some meaningful conclusions should be provided. Thus, the manuscript needs major revision prior to consider for publication.

Following suggestions may help to improve the quality of manuscript:

1. The “Machine learning date” in Fig. 1 was not covered in your manuscript, so is it suitable to put “Machine learning date” in the workflow schema of present study?

2. Please provide the thickness of the CFRP laminate in Figs. 2, 3 and 5.

3. Why 10.88 J was selected in the impact tests? In subsequent FEA, 0.5 to 10.0 J were designated as the impact energies for CFRP laminate, why not selected a higher energy (higher than 10.88 J in impact tests) such as 12 or 15 J?

4. It is better to explain how the material parameters and damage evolution parameters for CFRP laminate in Tables 1 to 3 are determined in detail. Additionally, in your FEA models, how were the various impact energies obtained, via changing the mass or velocity of the impactor? If the impactor velocity was adjusted, the strain rates of the CFRP laminate under impact loads may be different. However, the strain rate effect on those material parameters and damage evolution parameters was not considered.

5. How were the damage factors (d1, d2 and ds) determined and introduced in the post-impact FEA models? through UMAT of Abaqus?

6. In Table 1, the longitudinal elastic modulus (E1) and Poisson ratio (v1) was 129840 MPa and 0.30, respectively. However, the results in both tensile experiments and FEA showed that the values for E1 and v1 were 76965 MPa and 0.30, respectively. Please clarify the reasons for the discrepancy between the inputted elastic modulus and the tested one. Is it more reasonable to use the test results as the basic material parameters in Table 1?

7. It is better to display the impact damages of the bottom layer of the CFRP laminate in Fig. 6. In the common sense, it seems that when the impact energy was small, the damages on the layers closed to the impactor were severe, and the damages on the bottom layer may be lightest. However, the results in your FEA show that the damages in the top and bottom layer were most severe, a discussion of the impact damages for the bottom layer should be provided.

8. Do the layer stacking sequences affect the impact behavior of the entire CFRP laminate? If yes, which layer should be selected to evaluate the impact damages of the laminates with different layer orientations?

9. There is no doubt that the discrepancy of strains in the surroundings of impact damage of CFRP laminate exists. But how can the strains discrepancy be employed to quantify the impact damage in real applications?

10. It is better to provide some useful conclusions to enhance the general applicability of present study.

Author Response

Dear reviewer, we thank you for the comments that will improve the article. You can find responses to specific comments below. The manuscript has the same responses in bubbles, so you can easily see the change.

Comments

  • The “Machine learning date” in Fig. 1 was not covered in your manuscript, so is it suitable to put “Machine learning date” in the workflow schema of present study?

    Re: We agree that machine learning data is not the main topic of this paper, and it is not necessary to implement in workflow schema, so this point was deleted from Figure 1.

  • Please provide the thickness of the CFRP laminate in Figs. 2, 3 and 5.

Re: The thickness information was added in the text.

  • Why 10.88J was selected in the impact tests? In subsequent FEA, 0.5 to 10.0 J were designated as the impact energies for CFRP laminate, why not selected a higher energy (higher than 10.88 J in impact tests) such as 12 or 15 J?

Re: The impact energy about 10.88 J (10.45 J in average) was selected as reference, because the damage of material is already present however it is hardly detectable by visual control. Higher impact energy already causes visually detectable damage.

  • It is better to explain how the material parameters and damage evolution parameters for CFRP laminate in Tables 1 to 3 are determined in detail. Additionally, in your FEA models, how were the various impact energies obtained, via changing the mass or velocity of the impactor? If the impactor velocity was adjusted, the strain rates of the CFRP laminate under impact loads may be different. However, the strain rate effect on those material parameters and damage evolution parameters was not considered.

Re: As pointed bellow in answer of point 6, the parameters are only the products of optimization process.  The influence of speed and   of referenced in [9].

  • How were the damage factors (d1, d2and ds) determined and introduced in the post-impact FEA models? through UMAT of Abaqus?

Re: The damage properties are calculated following the validated process, which was presented in referenced paper [35]. Using this schema gives a chance to transfer the results between ABAQUS explicit and implicit solver and save lot of computational time. This feature is not commercially accessible.

  • In Table 1, the longitudinal elastic modulus (E1) and Poisson ratio (v1) was 129840 MPa and 0.30, respectively. However, the results in both tensile experiments and FEA showed that the values for E1and v1 were 76965 MPa and 0.30, respectively. Please clarify the reasons for the discrepancy between the inputted elastic modulus and the tested one. Is it more reasonable to use the test results as the basic material parameters in Table 1?

Re: The properties which shows Table 1 are the properties of one single layer (lamina). The properties of laminate, which consists of 12 differently oriented layers has different properties. They are shown in Table 7. and they were determined by experimental. The properties of one lamina were optimized by using the schema referenced in [9].

  • It is better to display the impact damages of the bottom layer of the CFRP laminate in Fig. 6. In the common sense, it seems that when the impact energy was small, the damages on the layers closed to the impactor were severe, and the damages on the bottom layer may be lightest. However, the results in your FEA show that the damages in the top and bottom layer were most severe, a discussion of the impact damages for the bottom layer should be provided.

Re: In the presented model, we consider the material as strain-rate independent, and influence of plasticity is neglected. To increase the calculation time of impact simulation with explicit solver the mass of impactor was increased. Nominal speed of impactor used with the presented model was 10 m/s. This increase of impactor speed does not have any influence on the response mode. That was also a reason tor keeping the laser detector under the impactor. The typical Abaqus explicit solver mass-scaling was rather not used. This phenomena was also studied in details, and referenced in paper [9] and [1].

  • Do the layer stacking sequences affect the impact behaviour of the entire CFRP laminate? If yes, which layer should be selected to evaluate the impact damages of the laminates with different layer orientations?

Re: Yes, the relative orientation angle affects the damage between the layers. This is shown in the Table 4. which is related to strength parameters of cohesive contact. But the damage level also depends on the position of the layer. From the results of simulations which is shown in the Figure 18 a) the most damaged is the 1st layer, which is placed in the opposite side of the impact. It is in balance with the experiments and referenced in [9].

  • There is no doubt that the discrepancy of strains in the surroundings of impact damage of CFRP laminate exists. But how can the strains discrepancy be employed to quantify the impact damage in real applications?

Re: In this paper we only assume the way of detection which is based on comparison with the reference (undamaged) state.

  • It is better to provide some useful conclusions to enhance the general applicability of present study.

Re: In this article, the material model of a composite laminate was shown. It’s parameters are fully sufficient for impact simulations with the help of the Abaqus/Explicit or Abaqus/Standard system. The credibility of the results was confronted with experiments.

But a specific example showed that the impact leads to higher damage on the side of the sample that is not exposed to the impact. On secondary elements of airframes, this kind of damage is usually not a problem. But it points to the special insidiousness of the phenomenon of poorly visible impact and the necessity of its identification on the first primary constructions. Although the damage from barely visible impact damage usually does not tend to spread, it can lead to a decrease in static stiffness and strength. The simulations further show that the damage rate is layer dependent in the laminate. It can be seen from the results that the failure of the layers in the vicinity of the mid-plane is lower than the failure rate on the surface.

It should be emphasized that the model results were only validated in the presented rate and damage of the laminate with a different composition and may be the same from those shown. In general, the material model can therefore be used as a reference for laminates with the same or very similar properties. The most important findings have been summarized in individual points shown below.